# Hydration of Nucleobase as Probed by Electron Emission of Uridine-5′-Mono-Phosphate (UMP) in Aqueous Solution Induced by Nitrogen K-Shell Ionization

**Yasuaki Takeda [1], Hiroyuki Shimada [1], Ryosuke Miura [1], Masatoshi Ukai [1,\*], Kentaro Fujii [2], Yoshihiro Fukuda [3,†] and Yuji Saitoh [3]**

[1]  Department of Applied Physics, Tokyo University of Agriculture and Technology, Koganei-shi, Tokyo 184-8588, Japan; saboten2828@gmail.com (Y.T.); shimada.hr.pf@gmail.com (H.S.); a_w-e.k_s@docomo.ne.jp (R.M.)

[2]  Center of Quantum Beam Science, National Institute for Quantum and Radiological Science (QST), Naka-gun, Ibaraki 319-1195, Japan; fujii.kentaro@qst.go.jp

[3]  Synchrotron Radiation Research Center, Japan Atomic Energy Agency (JAEA), Sayo-gun, Hyougo 679-5148, Japan; y-fukuda@spring8.or.jp (Y.F.); ysaitoh@spring8.or.jp (Y.S.)

\*  Correspondence: ukai3@cc.tuat.ac.jp; Tel.: +81-42-388-7222

†  Permanent address: SPring-8 Service Co. Ltd., Tatsuno, Hyogo 679-5165, Japan.

**Abstract:** To identify the precise early radiation processes of DNA lesions, we measure electron kinetic energy spectra emitted from uridine-5′ monophosphate (UMP) in aqueous solution for the photoionization of the N 1s orbital electron and for the following Auger effect using a monochromatic soft X-ray synchrotron radiation at energies above the nitrogen K-shell ionization threshold. The change of photoelectron spectra for UMP in aqueous solutions at different proton concentrations (pH = 7.5 and 11.3) is ascribed to the chemical shift of the N3 nitrogen atom in uracil moiety of canonical and deprotonated forms. The lowest double ionization potentials for aqueous UMP at different pH obtained from the Auger electron spectra following the N 1s photoionization values show the electrostatic aqueous interaction of uracil moiety of canonical (neutral) and deprotonated (negatively charged) forms with hydrated water molecules.

**Keywords:** DNA damage; UMP; synchrotron radiation; liquid microjet; electron spectroscopy

---

## 1. Introduction

DNA lesions induced by ionizing radiation can be the cause of cancers, cell deaths and mutations. In the induction of DNA lesions, the direct radiation effect via direct energy transfer from ionizing radiation to DNA has recently been revealed to be almost comparable to indirect effect [1–3], which are the result of reaction of DNA with diffusible reactive species [4–6], so that it is considered to be important. However, the precise molecular mechanisms from the primary energy donation to a specific site of DNA, such as to a nucleobase moiety, followed by the elementary physicochemical and chemical reaction processes of it to finally give rise to damage induction are not clarified yet.

In the radiation interaction with DNA, although the cross sections for outer shell excitation are large enough to absorb the energy of ionizing radiation efficiently, yet outer-shell excitation and ionization mainly induce single strand breaks of DNA at most, which are easily reparable by repair enzyme proteins. On the other hand, inner-shell excitation and ionization by the interaction of ionizing radiation can be the causes of the production of a few lesions among several base pairs in DNA, i.e.,

"clustered DNA damage", which is recognized to be non-detectable by repair enzymes so that is the cause of a significant biological effect [3]. The atomic site and element selective excitation and ionization of an inner shell orbital electron using monochromatic soft X-ray synchrotron radiation provides an opportunity to study the primary process of the direct effect by radiation. We have shown a peculiar enhancement of the formation of unstable radicals by the irradiation to DNA by the monochromatic soft X-rays at energies of the N 1s and O 1s thresholds [7]. We have also shown the selective excitation of the nucleobase moieties of nucleotides in aqueous solutions by measuring the X-ray absorption near-edge structure (XANES) spectra at energies in the vicinity of the nitrogen K-edge [8–10]. We also revealed aqueous interaction onto the biding energies of nitrogen K-shell orbital electrons of nucleobase moieties as a result of the structural change between the imine (double-bonded –N=) and amine (single-bonded –N<) bonding sites of N atom by the exchange of protons under hydration. We remark another aspect of hydration that the hydrated water molecules around DNA should suppress irreparable dissociation of doubly charged ions after the ionization of inner shell electron and following the Auger effect by absorbing and transferring the excess charge and internal energy to the cell liquid [11,12]. This is a role of hydration to allow ultrafast chemical processes under competition with interatomic (molecular) Coulomb decay [13]. A similar aspect of hydration is also suggested by the measurement of luminescence yields as a function of X-ray energy in the region between nitrogen and oxygen K-edges, which we have recently carried out [14]. However, the effect of hydration and aqueous interaction onto the ionization of inner shell electrons and the production of doubly charged ions are not known yet.

In this paper, we show the photoelectron spectrum and Auger electron spectrum of uridine-5′-monophosphate (UMP) in aqueous solution using a monochromatic soft X-ray synchrotron radiation at energies above the nitrogen K-shell ionization threshold under different pH. Although the electron spectrum measurements were carried out for water [15] and solvated proteins [16], those for DNA related molecules have not been published. We intend to confirm the effect of aqueous interaction to nucleobase moiety and identify the energy levels of single K-hole ions and outer shell doubly charged ions of uracil moiety of UMP under hydration. Those ions are the very early intermediates of reaction pathways which may lead to the direct radiation effect relating to the "clustered DNA damage".

## 2. Experimental

Commercially obtained disodium salt of UMP (Wako) was used without further purification. The solutions of 0.7-mol/L UMP at pH = 7.5 and 11.3 were prepared by dissolving the salts in purified water by a Milli-Q system (Millipore) and filtration through a 1 μm sterilizer to reduce aggregated reagents. The pH values of the solutions of UMP were adjusted by the addition of NaOH solution before filtration. [10].

Figure 1 (upper) shows the abundance of aqueous UMP structures calculated with the reference of acid dissociation constants p$K_a$ [17]. The phosphoric acid (PO$_4$) moiety is singly ionized at acidic solution, whereas the fully ionized PO$_4$ moiety is abundant in neutral (around pH = 7) and basic (pH > 10) solutions. Another characteristic difference arises at the uracil moiety (see Figure 1 (lower)). The uracil moiety of the aqueous UMP at around pH = 7 takes the canonical form, i.e., the N1 and N3 nitrogen atoms are both in amine (-N=) binding character, whereas the N3 nitrogen atom of uracil moiety is deprotonated at around pH = 11. The deprotonated N3 atom was shown to be of a similar character of imine (-N<) bond with respect to the electron density [10], so that there should exist two types of nitrogen atoms of different electronic properties in uracil moiety. The different structures of base moieties of aqueous nucleotides as well as aqueous UMP under the different pH were demonstrated by the spectral change of the N 1s XANES spectra measurements [9,10].

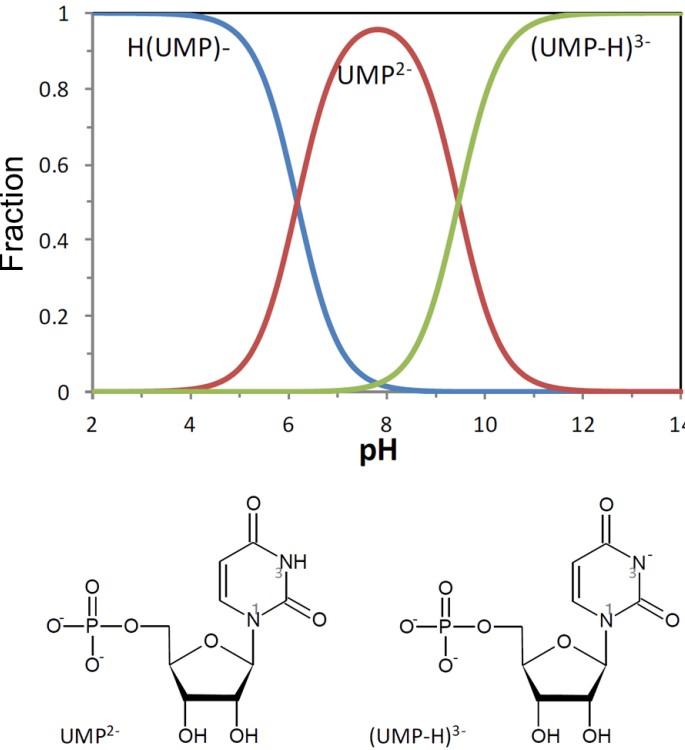

**Figure 1.** Fraction of aqueous uridine-5′ monophosphate (UMP) ions (**upper**) and their structures (**lower**). Uracil moiety of aqueous UMP in this experiment takes the canonical form (left, lower) at pH = 7.5 and the deprotonated form at the N3 atom (right, lower) at pH = 11.3 (see text).

Experiments were carried out by the irradiation to the aqueous UMP of the monochromatic soft X-rays at the BL-23SU beamline of the SPring-8 synchrotron radiation facility. The experimental setup and procedure were described elsewhere [8–10,15,18,19]. Briefly, an aqueous solution of a nucleotide in a form of liquid microjet is introduced into vacuum through a 20 μm diameter platinum orifice and is intersected by a focused beam of monochromatic soft X-ray synchrotron radiation. The electrons ejected from the intersection region enter an electron spectrometer through a 1 mm orifice and are dispersed with their energies using a hemispherical electrostatic analyzer [15]. The nominal energy width of the analyzer is 0.7 eV (fwhm). The absolute electron energy is calibrated by measuring the outermost valence electron spectra of water molecules in the liquid phase. The electron count rate at each soft X-ray energy is normalized for the photon intensity obtained from the drain current on the post-focusing mirror. The intensity of the soft X-ray in the energy range of the present measurements is typically around $10^{11}$ photons/s with a band pass of 0.38 eV (fwhm) at 400 eV photon energy. Calibration of the absolute photon energy is carried out for well-defined resonant transition energies of $N_2$ and $O_2$ molecules in the gas phase [20,21].

## 3. Calculation

For the identification of the photoelectron and Auger electron spectra of UMP, the ionization potentials of N 1s orbital electrons are calculated using the computational method based on the density functional theory (DFT). The main part of a DFT method was described in detail in the previous publication for the calculation of the XANES spectra for nucleotides [9,10], so that the outline only is depicted below. Based on the dominant molecular forms of the nucleotides in aqueous solutions [22], the calculations are made for their canonical and deprotonated forms (Figure 1). The program package StoBe-DeMon [23] is used for the calculation. Optimizations are carried out using the BE88 exchange functional [24] and the P86 correlation functional [25]. The triple-zeta valence plus polarization basis sets [26] for C, N, and O atoms and a contracted basis set for hydrogen atoms [23] are used.

In previous papers, we have shown the XANES spectra calculated in the vicinity of nitrogen K-shell edges for nucleotides [8–10]. Although the calculated XANES spectra were basically for the nucleobase moiety in the gas phase, they reproduced the corresponding experimental XANES spectra for nucleotides in solutions. The difference of XANES spectra in solutions of different pH values was systematically explained by the protonation or deprotonation structural change of purine- and pyrimidine-containing nucleotides at their nitrogen atomic sites under different pH atmosphere [10]. The reason for such a good systematic correspondence between the calculated XANES spectra for nucleobases in the gas phase and the experimental XANES for hydrated nucleotides may be due to the weak aqueous interaction with base moieties. We remark, however, the previous finding of protonation or deprotonation structural change under different pH atmosphere is the result of aqueous interactions with base moieties of donation or elimination of protons.

To calculate the ionization potentials of the N1 and N3 1s orbital electrons of UMP, we employ model molecules of 1-methyl uracil (referred to below as Meura), where N1 atom of the base is bound to a methyl group instead of to the C1' carbon atom of ribose group in UMP. It is reasonably expected that those model molecules share the electronic properties and structures in the base moieties to the corresponding nucleotides.

To take into the effect of aqueous interaction for the calculation of the double ionization potential of the ions at the uracil moiety in the model molecule of Meura, we employed the Gaussian09 package in combination with a polarizable continuum model (PCM) [27]. PCM approximates a fewfold hydration spheres directly surrounding uracil moiety and outer bulk water as a continuous and homogeneous insulator of the relative electrostatic permittivity of 78. The electron density distribution at the molecular sphere of the intact uracil moiety is influential to the net polarization of continuous water prior to irradiation. Thus, the optimized energies of interaction due mainly to the orientational polarization are not the same between the canonical and the deprotonated uracil moieties. Upon the prompt change of the charge state of the uracil moiety by the ionization of the N 1s orbital electron and following the Auger effect, the orientational polarization is assumed to remain stationary, so that only electronic polarization is optimized.

## 4. Results and Discussion

### 4.1. N 1s Photoelectron Spectra for UMP

Photoelectron spectra of the N 1s orbital electron of uracil moiety of UMP in aqueous solutions at different proton densities (pH = 7.5 and 11.3) are obtained. Figure 2 shows the X-ray photoelectron spectra (XPS) in the binding energy scale of UMP for solutions of pH = 7.5 and 11.3 at the photon energy of 520 eV. The energy widths of the XPS spectra are larger than the nominal widths of the incident monochromatic X-ray and the electron spectrometer due probably to the hydration to the uracil moieties [28]. However, the peak energies (center of gravity energies) of the photoelectron yields are close to the theoretical energy of the N 1s orbital(s) of UMP [10], which entails that these electrons are the result of the X-ray induced photoemission from the N 1s orbital(s) (N 1s XPS). However, the peak energies and the widths of the spectral envelops are somewhat different as follows. The peak energy of the N 1s XPS spectrum at pH = 7.5 locates at 404.7 eV, whereas the energy at pH = 11.3 locates at 403.5 eV. The width of the N 1s XPS spectrum at pH = 7.5 is 4.0 eV, whereas that for pH = 11.3 is 5.0 eV. This may be explained as follows. The nitrogen N1 and N3 atoms are both in the amine character at pH = 7.5 so that the binding energies of the N1 1s and the N3 1s orbital electrons are essentially the same. On the other hand, the N3 at pH = 11.3 is in deprotonated imine character, so that the binding energies of the N1 1s and the N3 1s orbital electrons are split. The difference of binding energies was shown in the XANES spectra of UMP in the vicinity of nitrogen K-edge as follows; the excitation energy of the N3 1s electron to the lowest unoccupied $\pi^*$ orbital for the deprotonated uracil moiety of aqueous UMP at pH = 11.3 was smaller by about 2 eV in comparison with that for the canonical uracil moiety at pH = 7.5. This observation was qualitatively in agreement with the results

of theoretical transition energies for canonical and deprotonated UMP in isolated (not aqueous) forms calculated by the DFT method.

However, we recognize a discrepancy between the absolute peak energies in the present N 1s XPS spectra for aqueous UMP and the calculated ionization potentials (IPs) for the N1 1s and N3 1s orbital electrons by the DFT method. As shown in Figure 2, the peak energy of the N 1s XPS spectrum for UMP in a solution of pH = 7.5 locates at 404.7 eV, whereas the energy in a solution of pH = 11.3 locates at 403.5 eV. The calculated IPs of the N1 1s and the N3 1s orbital electrons for the canonical Meura are shown to be almost degenerate and located at around 406 eV, which is relatively close to the peak energy of the N 1s XPS spectrum at pH = 7.5. On the other hand, the calculated IPs of the N1 1s and the N3 1s orbital electrons for the deprotonated Meura are not degenerate, i.e., the IP of the N1 1s orbital electron is at 401.0 eV and the IP of the N3 1s orbital electron is at 398.2 eV. The difference of the calculated IPs for different forms of uracil moieties is much larger than that of the experimental peak energies, which may be ascribed to the difference in the effect of the aqueous interaction or hydration to uracil moieties as described below again for the analysis of the Auger electron spectra of UMP in solutions of different pH. Table 1 summarizes the observed and calculated results of N 1s ionization potentials.

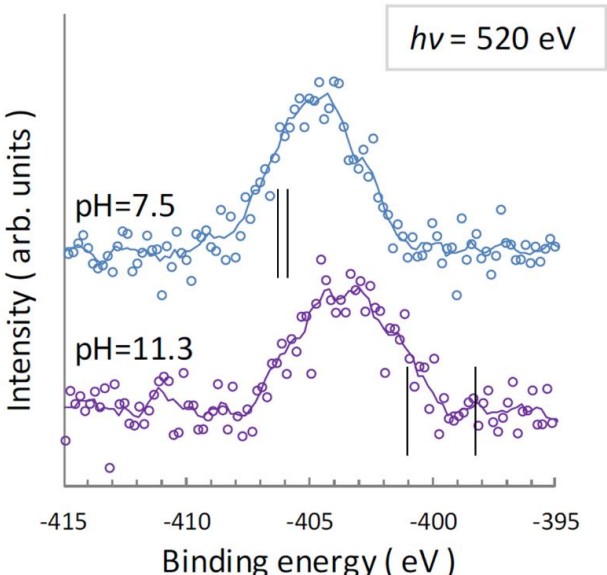

**Figure 2.** X-ray photoelectron spectra of UMP for solutions of pH = 7.5 and 11.3 at the photon energy of 520 eV. Vertical bars indicate the theoretical ionization potentials (IPs) of the N1 1s and N3 1s orbitals (see text).

**Table 1.** Observed and calculated N 1s ionization potentials for aqueous uracil moiety. The observed results are for aqueous UMP in XPS at a photon energy of 520 eV and the calculated results for comparison are for isolated Meura (see text).

|  | pH = 7.5 | pH = 11.3 | |
|---|---|---|---|
|  | **N1, N3 (Mixed)** | **N1** | **N3** |
| Observed [a] | 404.7 eV, 404.9 eV | 404.9 eV | 402.5 eV |
| Center of Gravity | 404.7 eV | 403.5 eV | |
| Calculated [b] | 405.9 eV, 406.3 eV | 401.0 eV | 398.2 eV |

[a] Obtained by a doubly gaussian fit. [b] DFT calculation (see text).

### 4.2. Auger Electron Spectra of Aqueous UMP Following N 1s Photoionization

Figure 3 shows electron yields as a function of the kinetic energy between 338 eV and 400 eV for UMP in a solution of pH = 7.5 and the liquid water at the X-ray photon energy of 490 eV. To remove the effect of electrification of liquid water, the measurement is carried out for dilute (0.1 mol/L) NaCl solution. Since, however, sodium and chloride ions do not emit electrons of characteristic energy in this kinetic energy region, the spectrum for the NaCl solution is safely regarded as that to "pure liquid water". The electron yields for the "pure liquid water" decrease monotonously with the decrease of electron kinetic energy, the behavior of which can be fitted by a solid curve in Figure 3. The electron yields for aqueous UMP in a solution at pH = 7.5 is almost identical to those for "pure liquid water" in the kinetic energy region from 385 eV to 400 eV, so that the common yield spectra in this region are considered to be the background contribution due to the ionization of liquid water. The yield spectrum for the aqueous UMP in the kinetic energy region from 338 eV to 385 eV shows the characteristic feature due to the ionization and Auger process among the aqueous UMP, which can be regarded to be based on the background electron yields of the "pure liquid water" as approximated by the solid curve. Thus, the electron yields for the aqueous UMP are subtracted by the background yields due to the "pure liquid water" and then normalized for the photon flux as shown in Figure 4.

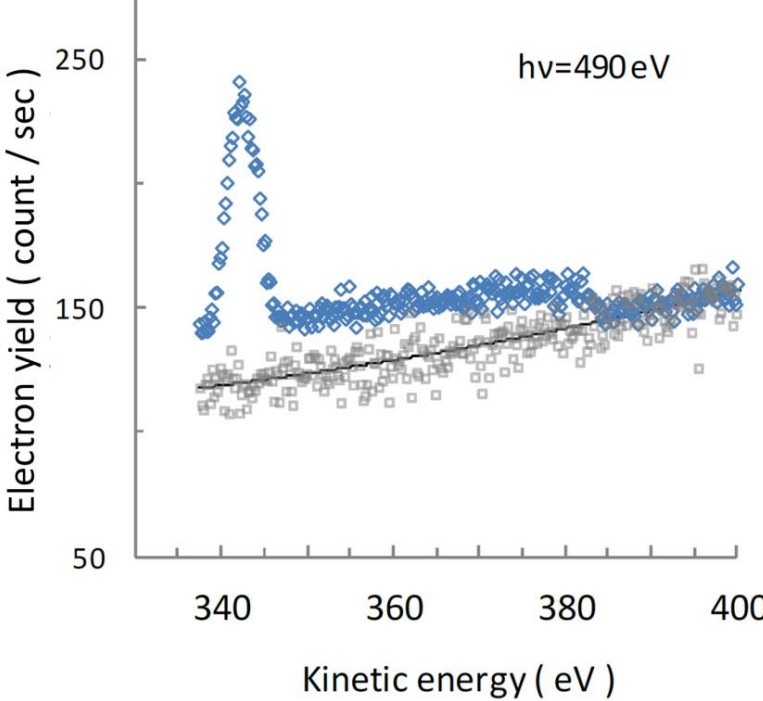

**Figure 3.** Electron kinetic energy spectra for aqueous UMP at pH = 7.5 (◇) and the liquid water (□) at the X-ray photon energy of 490 eV.

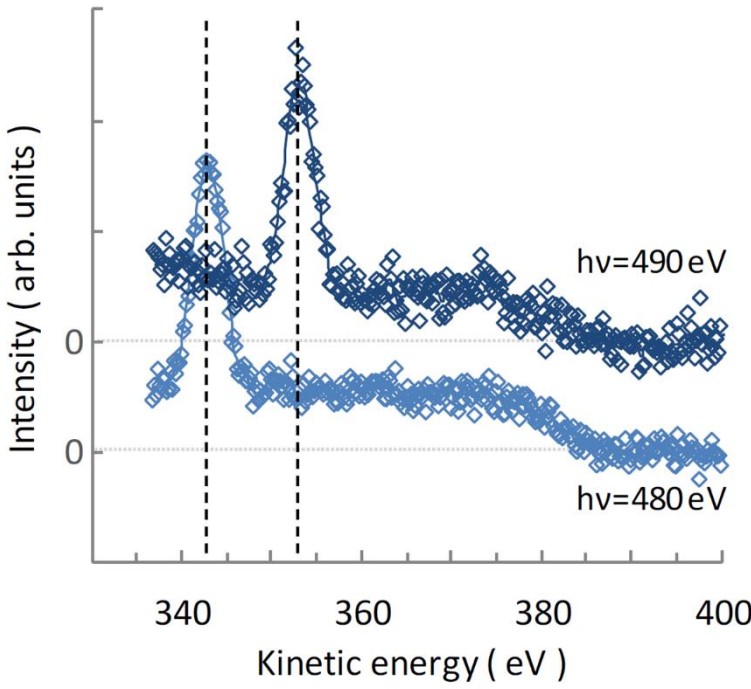

**Figure 4.** Electron kinetic energy spectra for aqueous UMP at pH = 7.5 at the X-ray photon energies of 480 eV and 490 eV.

Figure 4 shows the electron yields for the ionization of the aqueous UMP obtained at the photon energies of 480 eV and 490 eV after the subtraction of the background yields due to "pure liquid water". It is obvious that these spectra of electron yields are comprised of two components, i.e., strongly enhanced yields presenting a peak structure at the smaller kinetic energy region and continuously enhanced yields at kinetic energies below ca. 385 eV. As seen, the kinetic energy of the peak maximum of the strongly enhanced yields at the photon energy of 480 eV is 343 eV, whereas the kinetic energy at the photon energy of 490 eV is 353 eV. The strongly enhanced yields are thus explained due to the photoionization of the P 2p orbital electrons in the phosphoric acid moiety, the IP of which is about 140 eV [25]. On the other hand, the continuously enhanced yields of electrons stationary staying below 395 eV at different photon energies are explained as the Auger electrons emitted following the photoionization of the N 1s orbital electrons, the IP of which is about 400 eV as discussed above. This is due to the following reasons [29]. (1) The IP of the O 1s orbital electron (about 530 eV) cannot be accessed with the photon energy of 480 eV or 490 eV. (2) The electron emission associated with the C 1s orbital electron (IP about 290 eV), such as its photoionization and Auger process, cannot emit an electron with the kinetic energy in the region of 340 eV–395 eV. (3) The photoionization of an outer valence electron of UMP produces an electron with much larger kinetic energy. This discussion is also the case for the kinetic energy spectra of electron yields for aqueous UMP at pH = 11.3.

Figure 5 shows the N 1s Auger electron spectra derived as described above for aqueous UMP at pH = 7.5 and 11.3 at the photon energy of 480 eV. The spectra are tilted by a right angle to make them correspond to the energy level diagram, i.e., the electron yields are presented by abscissa into the right and the ordinate is the energy scale (vertically increasing) of the double ionization potential (DIP) for di-cations of uracil moiety in the aqueous UMP. Although in the previous Auger electron spectra for gaseous molecules distinct peak enhancements of electron yields were shown to indicate a number of electronic states of di-cations [30], Figure 5 only shows continuously enhanced electron yields. The reason may be ascribed to the broadening by hydration of the IPs of N 1s orbital and DIPs of di-cations, but is not identified clearly at present. So we restrict the following discussion to the lowest values of DIPs.

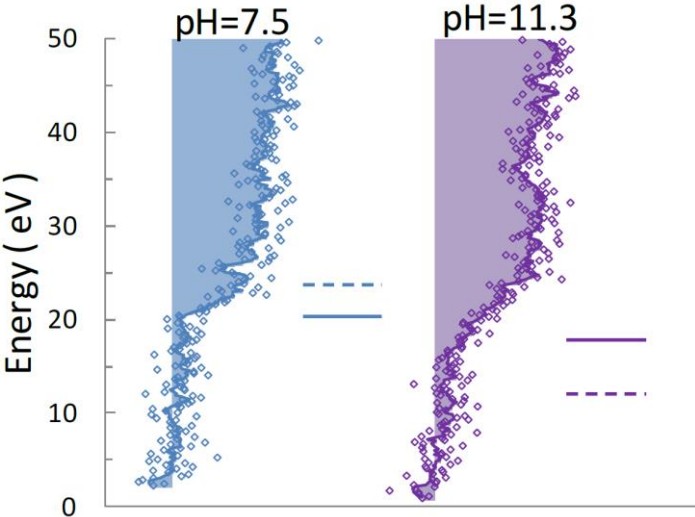

**Figure 5.** Electron spectra for aqueous UMP at pH = 7.5 and pH = 11.3 in the ionization energy of di-cations. For comparison, the theoretical double ionization potentials (DIPs) are shown by dashed and solid horizontal lines for isolated and hydrated Meura (model molecule for UMP), respectively (see text).

The values of DIP are obtained by a conversion using the relation, KE = IP(N1s)- DIP, where KE is the kinetic energy in the spectra such as Figure 4, IP(N1s), the IP for the N 1s orbital electron. The employed IP(N1s)s are the experimentally obtained peak energies (center of gravity energies) of the photoelectron yields as shown in Figure 2, i.e., 404.7 eV for the Auger spectrum at pH = 7.5 and 403.5 eV, at pH = 11.3. Both Auger electron spectra show the enhancement of electron yields at about 20 eV to which the lowest DIPs should correspond. The appearance energy (lowest DIP) seems to be slightly smaller for the deprotonated form of uracil moiety at pH = 11.3. Precise peaks of Auger electron yields are absent, which may be ascribed to the broadening of individual Auger electron peaks due to aqueous interaction of the uracil moieties. Broadening due to inelastic collisions of Auger electrons with water molecules may presumably be included for larger DIPs in these spectra.

To certify that the appearance energies of the enhancement of the Auger electron yields correspond to the lowest DIPs, the calculated lowest DIPs of the canonical uracil moiety (corresponding to pH = 7.5) and the deprotonated uracil moiety at N3 (corresponding to pH = 11.3), both for isolated (not aqueous) Meura using the DFT method, are shown by dashed horizontal lines. The agreement of the appearance energies of the enhancement of electron yields with the theoretical lowest DIPs is not favorable at pH = 7.5 or pH = 11.3, which seems to be of a similar behavior of the agreement of the IPs of the N 1s orbital electron shown in Figure 2. We are thus obliged to consider the limitation of the calculated IPs and DIPs using the molecules in the isolated (not aqueous) model molecule of Meura, so that other DIPs are calculated using the DFT method including a polarizable continuum model (PCM) [27], as described above in the calculation section, shown by the solid horizontal lines in Figure 5. A significantly favorable agreement of the appearance energies of the enhancement of electron yields with the theoretical lowest DIPs including PCM is obtained for both results at pH = 7.5 or pH = 11.3. The agreement thus obtained can be ascribed as follows. The canonical uracil moiety expected in a solution at pH = 7.5 prior to the ionization of the N 1s orbital electron is neutral in charge, whereas the moiety in a solution at pH = 11.3 is negatively charged due to deprotonation, so that the optimized orientational polarization of the aqueous interaction or hydration is different. Upon the photoionization of the N 1s orbital electrons and following Auger processes, the charge state of the canonical (pH = 7.5) uracil moiety becomes doubly charged, whereas the state of the deprotonated moiety becomes singly charged. Since the photoionization and Auger process are prompt, the orientational polarization of aqueous interactions stays frozen for both intact structures. As a result, the DIP of the uracil moiety in the canonical form is slightly reduced by the electronic polarization (about $\varepsilon_r = 2$) of surrounding

water molecules. On the contrary, the DIP of the uracil moiety in the deprotonated form is increased due to the coulombic repulsion of the surrounding positive charge density of water molecules which keep the intact orientational polarization (about $\varepsilon_r = 78$) to the positively charged uracil moiety. Such an effect of orientational polarization is also the case for single ionization of the N 1s orbital electrons at pH = 7.5 and pH = 11.3 observed as the discrepancies of the IPs in the XPS spectra to the theoretical IPs calculated for isolated (not aqueous) forms of Meura.

**Author Contributions:** M.U. wrote the manuscript. M.U., H.S., K.F., Y.F. and Y.S., developed the experimental apparatus. Y.T. and H.S. developed the method of calculation. U.M., H.S., Y.T., R.M., and K.F. performed the experiments and the data analysis. M.U. supervised the research. All authors have read and agreed to the published version of the manuscript.

**Funding:** This work was supported by financially from JSPS KAKENHI (Grant Nos. 21241017 and 25241010).

**Acknowledgments:** The experiments at SPring-8 were carried out under approval of the Japan Synchrotron Radiation Research Institute (JASRI) Proposal Review Committee for Proposal Nos. 2013B3810, 2014A3810, 2014B3810, 2015A3810 and 2015B3810.

**Conflicts of Interest:** The authors declare no known conflict of interest.

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
