# Peer review of "Hydration of Nucleobase as Probed by Electron Emission of Uridine-5′-Mono-Phosphate (UMP) in Aqueous Solution Induced by Nitrogen K-Shell Ionization"

_qubs, doi:10.3390/qubs4010010_

Round 1

Reviewer 1 Report

The authors present a study investigating the electron spectra of core ionized solvated UMP in a liquid jet environment. The experiment was conducted carefully and the results are promising. However, there are mayor concerns about the quality of the manuscript. 

A) Foremost, the mistreatment of the English language makes it on several occasions impossible to conceive the information. As it is a very interesting direction of research I am willing to give an in-depth feedback to the manuscript. Nevertheless, it is of utmost importance that the whole manuscript is revisited for spelling and even more important readability. 

B) minor issues

L33: DNA is recently revealed -> DNA has recently been revealed

L40: induce single strand breaks

L41: proteins.

L43: by repair enzymes

L48: N1s and O1s

L79: the fully ionized

L80: the uracil moiety

L84: Shimada, 2015 is reference 8?

L107: so that the outline only is depicted below

L108: of the nucleotide

L114: XANES spectra calculated

L116: gas phase they reproduced the

L120: the reason for

L127: to a methyl group

L137: why does the permittivity has no unit?

L139: remain stationary 

L145: 520(.)eV

L145: spectra are larger than

L150: of the spectra envelops

L166: Meura -> (my best guess is) moiety, also L169, L249, L264

L200: obvious that these spectra

L201: a peak structure at

L203: peak maximum of the strongly enhanced

L224: (N1s) - DIP

L236: Figure 5: Electron spectra

L249 - L250: is obtained for both results at pH = 7.5 and pH=11.3.

L255: becomes doubly charged

L257: the DIP of the uracil 

L258 - L259: of the surrounding water molecules. On the contrary, 

L262: orientational polarization

C) questions

L64: which energy is to be identified? Do you mean to quantify the binding energy?

L124: what are photon ions? 

L134 - 137: I am not able to comprehend the very long sentence. Please rephrase that.  

L145-149: The sentence is too long. Please truncate it. 

L195: Figure 3: in the text a normalization to the photon flux is mentioned, nevertheless the intensity is given in counts/sec.

L216: Figure 4: why is the Auger region completely without any structure?Typically, Auger spectra close to the threshold show a wealth of structure.

L227 - L229: I am not able to comprehend the "should be indicated to locate" part. 

L254 - 257: please rephrase this very long sentence. 

Author Response

Thank you very much for your good evaluation of our paper. We have sincerely appreciated your efforts of critical reading of the manuscript.

We have understood that your comments B) and questions C) are the practical pointing out to the overall comment A) for the shortcoming of English in ouor manuscript;

A) Foremost, the mistreatment of the English language makes it on several occasions impossible to conceive the information. As it is a very interesting direction of research I am willing to give an in-depth feedback to the manuscript. Nevertheless, it is of utmost importance that the whole manuscript is revisited for spelling and even more important readability.

So according to your revision requests of B) we have revised the manuscript. In the way we have also recognized a number of other mistakes in English, they all have been mended. Every changes are shown in Track Change windows.

B) minor issues

L33: DNA is recently revealed -> DNA has recently been revealed

L40: induce single strand breaks

L41: proteins.

L43: by repair enzymes

L48: N1s and O1s

L79: the fully ionized

L80: the uracil moiety

L84: Shimada, 2015 is reference 8?

L107: so that the outline only is depicted below

L108: of the nucleotide

L114: XANES spectra calculated

L116: gas phase they reproduced the

L120: the reason for

L127: to a methyl group                                                                 

L139: remain stationary 

L145: 520(.)eV

L145: spectra are larger than           

L150: of the spectra envelops

L200: obvious that these spectra

L201: a peak structure at

L203: peak maximum of the strongly enhanced

L224: (N1s) - DIP

L236: Figure 5: Electron spectra

L249 - L250: is obtained for both results at pH = 7.5 and pH=11.3.

L255: becomes doubly charged

L257: the DIP of the uracil 

L258 - L259: of the surrounding water molecules. On the contrary, 

L262: orientational polarization                                                           

On the point of

“L137: why does the permittivity has no unit”,

since, corresponding sentence has been deleted by the edition, so answer is shown in Track Change window as “this is relative permittivity.”

On the point of

“L166: Meura -> (my best guess is) moiety, also L169, L249, L264”

As shown L126, Meura is not a moiety in nucleotide, but a model molecule (1-methyl uracil)..

C) questions

On “L64: which energy is to be identified? Do you mean to quantify the binding energy?”:

This is rephrased as “identify the energies of single K-hole ions and outer shell doubly charged ions of uracil moiety of UMP under hydration”.

On “L124: what are photon ions?”:

This is corrected as photon ions→proton ions

On, “L134 - 137: I am not able to comprehend the very long sentence. Please rephrase that,”  

“L145-149: The sentence is too long. Please truncate it,” 

and “L254 - 257: please rephrase this very long sentence,” 

These long sentences were rephrased into two sentences.

On “L195: Figure 3: in the text a normalization to the photon flux is mentioned, nevertheless the intensity is given in counts/sec.”:

  Yes, you are right. Figure 3 was raw data. We inserted appropriate explanation to derive the spectra from the raw data.

On “L216: Figure 4: why is the Auger region completely without any structure? Typically, Auger spectra close to the threshold show a wealth of structure.”:

The reason is not identified but your pointing out comment is appropriately treated in the text. We have added a typical reference of Auger spectra.

On “L227 - L229: I am not able to comprehend the "should be indicated to locate" part.”:

This is corrected. 

Reviewer 2 Report

This is a good work that could be published in QBS. However, there are some minor issues to be addressed.

Page 5: The authors need to provide a clear explanation of "a discrepancy between the absolute peak energies in the present N 1s XPS spectra for aqueous UMP and the calculated ionization potentials (IP’s) for the N1 1s and N3 1s orbital electrons by the DFT method".  A table containing the IPs of N1 and N3 1s orbital electrons of UMP at various pH and 1-methyl uracil by theory and the XPS peaks have to presented.

Author Response

Thank you very much for your good evaluation of our paper. Thank you for your recommendation of minor revision.

According to your recommendation of we have presented Table I containing the IPs of N1 and N3 1s orbital electrons of UMP at two pH values of solution.

Round 2

Reviewer 1 Report

Thank you for your clarifications. The manuscript makes more sense to me now.

There are still a few open questions:

1)

You write now: “We intend to confirm the effect of aqueous interaction to nucleobase moiety and identify the energies of single K-hole ions and outer shell doubly charged ions of uracil moiety of UMP under hydration.”

From this sentence it is still not clear what energies you are addressing.

My guess is that you mean: “[…] identify the energy levels of single K-hole ions and […]

2)

You twice write proton ions now. But protons are, by definition, hydrogen ions. Just delete the ions in both occasions.

Some additional (more general) remarks:

A) The quality of the results figures is poor. Most are pixelated and not easy to read.

B) The paper is readable but not well structured. It seems to me that the paper is mainly a release of data, which is not problematic in itself, but the introduction suggest that radiation damage is addressed within the paper.

C) The introduction lacks profoundly in the representation of work performed by other groups. This should include reviews on radiations damage (e.g. E. Alizadeh, L. Sanche, Precursors of solvated electrons in radiobiological physics and chemistry. Chemical reviews. 112, 5578–5602 (2012), doi:10.1021/cr300063r, C. Garrett et al., Role of water in electron-initiated processes and radical chemistry. Chemical reviews. 105, 355–390 (2005), doi:10.1021/cr030453x) and electron spectra of other solvated proteins (e.g. N. Ottosson, M. Faubel, S. E. Bradforth, P. Jungwirth, B. Winter, Photoelectron spectroscopy of liquid water and aqueous solution. Journal of Electron Spectroscopy and Related Phenomena. 177, 60–70 (2010), doi:10.1016/j.elspec.2009.08.007, A. Yokoya, T. Ito, Photon-induced Auger effect in biological systems. International journal of radiation biology. 93, 743–756 (2017), doi:10.1080/09553002.2017.1312670). To me it seems that the authors are focused too much on their own work. Nowhere it is discussed that the electronic states addressed in this paper are discussed heavily to be precursors for ultra-fast intermolecular processes (e.g. C. Richter et al., Competition between proton transfer and intermolecular Coulombic decay in water. Nature communications. 9, 4988 (2018), doi:10.1038/s41467-018-07501-6 or M. Mucke et al., A hitherto unrecognized source of low-energy electrons in water. Nature Phys. 6, 143–146 (2010), doi:10.1038/nphys1500).

D) I am not sure, and I was also not asked during the review process, whether the scope of the journal is appropriate for the research. I guess it is part of a special issue otherwise I would suggest other journals more specific for the presented research.

Author Response

Dear Reviewer,

Sincerely yours, Masatoshi Ukai    
